# Secreted CLCA1 modulates TMEM16A to activate Ca²⁺-dependent chloride currents in human cells

Monica Sala-Rabanal[1,2†], Zeynep Yurtsever[2,3,4,5†], Colin G Nichols[1,2], Tom J Brett[1,2,4,5,6]*

[1]Department of Cell Biology and Physiology, Washington University School of Medicine, St Louis, United States; [2]Center for the Investigation of Membrane Excitability Diseases, Washington University School of Medicine, St Louis, United States; [3]Biochemistry Program, Washington University School of Medicine, St Louis, United States; [4]Department of Internal Medicine, Washington University School of Medicine, St Louis, United States; [5]Drug Discovery Program in Pulmonary and Critical Care Medicine, Washington University School of Medicine, St Louis, United States; [6]Department of Biochemistry and Molecular Biophysics, Washington University School of Medicine, St Louis, United States

**Abstract** Calcium-activated chloride channel regulator 1 (CLCA1) activates calcium-dependent chloride currents; neither the target, nor mechanism, is known. We demonstrate that secreted CLCA1 activates calcium-dependent chloride currents in HEK293T cells in a paracrine fashion, and endogenous TMEM16A/Anoctamin1 conducts the currents. Exposure to exogenous CLCA1 increases cell surface levels of TMEM16A and cellular binding experiments indicate CLCA1 engages TMEM16A on the surface of these cells. Altogether, our data suggest that CLCA1 stabilizes TMEM16A on the cell surface, thus increasing surface expression, which results in increased calcium-dependent chloride currents. Our results identify the first Cl⁻ channel target of the CLCA family of proteins and establish CLCA1 as the first secreted direct modifier of TMEM16A activity, delineating a unique mechanism to increase currents. These results suggest cooperative roles for CLCA and TMEM16 proteins in influencing the physiology of multiple tissues, and the pathology of multiple diseases, including asthma, COPD, cystic fibrosis, and certain cancers.

*For correspondence: tbrett@wustl.edu

†These authors contributed equally to this work

Competing interests: The authors declare that no competing interests exist.

## Introduction

The calcium-activated chloride channel regulator (CLCA- previously known as chloride channel calcium activated) proteins (*Cunningham et al., 1995*) are a family of secreted self-cleaving metalloproteases that activate calcium-dependent chloride currents ($I_{CaCC}$) in mammalian cells (*Yurtsever et al., 2012*). CLCA family members are highly expressed in mucosal epithelia where they play important roles in mucus homeostasis and related diseases (*Patel et al., 2009*). For example, human CLCA1 plays a central role in interleukin (IL-) 13-induced mucus cell metaplasia, the main source of inflammatory mucus overproduction in chronic obstructive airway diseases, such as asthma and COPD (*Alevy et al., 2012*). Both clinical and animal model studies suggest a compensatory role for CLCAs in the context of cystic fibrosis (CF): the fatal intestinal disease, meconium ileus, arising in CFTR-deficient mice is corrected by overexpression of mCLCA3 (an orthologue of human CLCA1) (*Young et al., 2007*) and, correspondingly, mutations in *CLCA1* are found in a subset of CF patients with aggravated intestinal disease (*van der Doef et al., 2010*). At the cellular level, overexpression of

**eLife digest** Many biological processes that are important for our health involve the movement of ions into, and out of, our cells. For example, the flow of chloride ions out of cells controls the production of the sticky mucus that lines our windpipe and other airways. This mucus helps trap pollution and other foreign particles before they reach our lungs, and thus protects the lungs from harm. However in some diseases—such as cystic fibrosis and asthma—excessive amounts of thick mucus are produced; this can lead to breathing difficulties and an increased risk of infection.

Proteins belonging to the CLCA protein family were first thought to act as channels that allow chloride ions to flow through cell membranes. Later studies then revealed that these proteins are not channels; instead they trigger the movement of chloride ions across cell membranes by activating other channel proteins. However, the identity of these channel proteins was unknown, and it was unclear how CLCA proteins might activate these channels.

Sala-Rabanal, Yurtsever et al. have now shown that a member of the CLCA protein family, called CLCA1, is released from human cells and causes nearby cells to release chloride ions when the channel detects calcium ions. The movement of chloride ions triggered by CLCA1 looked very similar to the way chloride ions flow through a channel protein called TMEM16A, and so Sala-Rabanal, Yurtsever et al. asked whether these two proteins interact.

TMEM16A was discovered several years ago, but remains the only calcium-dependent chloride channel known in mammals. Sala-Rabanal, Yurtsever et al. showed that adding CLCA1 to cells caused more TMEM16A channels to appear in the cell surface membrane and thereby increased the flow of chloride ions. The CLCA protein also physically interacted with the chloride channel in the membrane to stabilize it; no other protein has been shown to regulate ion channels in this way before.

The findings of Sala-Rabanal, Yurtsever et al. provide a much clearer understanding of how the CLCA protein and the chloride channel work. Both of these proteins are known to contribute to excess mucus production in airway diseases; and both have been linked to cardiovascular diseases and certain cancers. These new findings may therefore also help researchers to target these proteins and develop treatments for these diseases.

---

CLCA proteins leads to activation of calcium-dependent chloride currents (*Gandhi et al., 1998*; *Britton et al., 2002*; *Elble et al., 2002*; *Greenwood et al., 2002*), and this functional observation had caused CLCAs to be initially misidentified as calcium-activated chloride channels (CaCCs) themselves (*Cunningham et al., 1995*). However, further bioinformatic and biochemical studies have demonstrated that CLCA proteins are secreted, soluble proteins and that they act to modulate CaCCs that are endogenous to mammalian cells (*Gibson et al., 2005*; *Hamann et al., 2009*; *Yurtsever et al., 2012*). The molecular identity of these channels, the mechanism of CLCA activation, and their potential roles in CLCA-mediated diseases, remain unknown.

TMEM16A (also known as Anoctamin1/DOG1) was recently identified as the first genuine CaCC in mammals by three independent groups (*Caputo et al., 2008*; *Schroeder et al., 2008*; *Yang et al., 2008*). 10 members of the TMEM16/Anoctamin family have been identified (TMEM16A-K, or Ano1-10); these proteins, predicted to be transmembrane proteins with eight membrane-spanning helices, have been found to function predominantly as CaCCs (TMEM16A and B) or as phospholipid scramblases (TMEM16C, D, F, G, and J) (*Pedemonte and Galietta, 2014*). TMEM16A, the best-characterized member of the family to date, is expressed in airway epithelia and smooth muscle, and its activity recapitulates some of the airway disease traits associated with CLCA1. Not only is TMEM16A expression significantly increased by IL-13 and IL-4 in primary cell models of chronic inflammatory airway disease (*Caputo et al., 2008*; *Alevy et al., 2012*), but TMEM16A overexpression is also linked to mucus cell metaplasia and airway hyperreactivity (*Huang et al., 2012*; *Scudieri et al., 2012*). In addition, TMEM16A-specific inhibitors decrease mucus secretion and airway hyperreactivity in cellular models (*Huang et al., 2012*). Although experiments with purified TMEM16A protein reconstituted in liposomes indicate that it can form a functional channel on its own (*Terashima et al., 2013*), several cytosolic modulators and interaction partners, such as calmodulin, phosphatidylinositol 4,5-bisphosphate ($PIP_2$), ezrin, radixin, and moesin, have been described (*Tian et al., 2011*; *Perez-Cornejo et al., 2012*; *Pritchard et al., 2014*). However, no

secreted regulators of TMEM16A activity have been identified as of yet. Here we report that secreted CLCA1 modulates TMEM16A-dependent calcium-activated chloride currents, and that this activation can occur in a paracrine fashion. Furthermore, we show that CLCA1 and TMEM16A co-localize and physically interact on the surface of mammalian cells, and that CLCA1 increases the level of TMEM16A protein at the cell surface, representing a novel mechanism of channel regulation by a secreted protein. We thus demonstrate a first downstream target of CLCA proteins and provide the first example of a secreted protein modulator of TMEM16A activity. These findings have significant implications for the roles of CLCA1 and TMEM16A proteins as cooperative partners, not only in the physiology and pathophysiology of the airways, but also in those of other tissues and organs.

## Results

### Secreted CLCA1 can activate Ca²⁺-dependent chloride currents in a paracrine fashion

We previously demonstrated that $I_{CaCC}$ are activated in HEK293T (293T) cells overexpressing human CLCA1 (*Yurtsever et al., 2012*). Given that CLCA1 proteins are cleaved and secreted from these cells, we hypothesized that exogenous CLCA1 may activate $I_{CaCC}$. In a first set of experiments to test this idea, GFP-expressing cells that had been co-cultured overnight with cells transfected with CLCA1-pHLsec plasmid (CLCA1) or with empty pHLsec vector (pHLsec) were tested for $I_{CaCC}$ by means of whole-cell patch clamp electrophysiology (*Figure 1A*). In the presence of 10 μM intracellular Ca²⁺ and physiological concentrations of extracellular Cl⁻, robust, slightly outward rectifying currents were activated in cells co-cultured with CLCA1-transfected cells, but only substantially smaller currents were detected in cells co-cultured with vector-transfected cells (*Figure 1B–D*). In a complementary experiment, whole-cell $I_{CaCC}$ were measured in untransfected cells that had been cultured in medium obtained from CLCA1- or from pHLsec-transfected cells (*Figure 2A*). We observed activation of large currents in cells exposed to CLCA1-conditioned medium that had the same Ca²⁺- and voltage-dependence properties as those induced in cells co-cultured with CLCA1-expressing cells (*Figure 2B–D*). As shown in *Figure 2B–C*, outward rectification of the CLCA1-activated current decreases at higher Ca²⁺ concentrations. In addition, current reversal potential shifts positive upon lowering extracellular Cl⁻. These features are in agreement with the properties of the Ca²⁺-dependent Cl⁻ conductance observed in CLCA1-expressing 293T cells (*Hamann et al., 2009*; *Yurtsever et al., 2012*), and consistent with those of CaCCs in native cells and heterologous expression systems (*Jeong et al., 2005*; *Yamazaki et al., 2005*; *Xiao et al., 2011*). These data indicate that secreted CLCA1 can activate $I_{CaCC}$ in a paracrine fashion.

### CLCA1-dependent $I_{CaCC}$ are carried by TMEM16A

We next focused on identifying the CaCC responsible for carrying the CLCA1-mediated currents. The CLCA1-modulated $I_{CaCC}$ in 293T cells are Ca²⁺-dependent, moderately outward rectifying in the presence of μM concentrations of intracellular Ca²⁺, Cl⁻-selective, and blocked by gluconate (*Hamann et al., 2009*; *Yurtsever et al., 2012*) (*Figures 1, 2*), closely resembling the biophysical characteristics of those observed for TMEM16A currents in heterologous expression systems (*Schroeder et al., 2008*; *Yang et al., 2008*; *Xiao et al., 2011*), proteoliposomes (*Terashima et al., 2013*), and native tissues (*Caputo et al., 2008*).

Given the biophysical and pathophysiological parallels between TMEM16A currents, and those activated by CLCA1, we hypothesized that CLCA1-activated currents may be carried by TMEM16A. Consistent with this idea, 293T cells were transfected with either TMEM16A siRNA or with non-specific, scrambled RNA (siControl), and cultured in CLCA1-conditioned medium. Exposure to secreted CLCA1 led to the activation of $I_{CaCC}$ in siControl-transfected cells (*Figure 3A,B*) that were comparable to the activation recorded in untransfected cells (*Figure 2B,D*), but these CLCA1-dependent currents were knocked down to essentially background levels in TMEM16A siRNA-transfected cells (*Figure 3A,B*). The TMEM16A siRNA significantly decreased expression of TMEM16A protein, assessed by Western blot (*Figure 3C*). These results demonstrate that CLCA1-dependent $I_{CaCC}$ in 293T cells are indeed carried by TMEM16A.

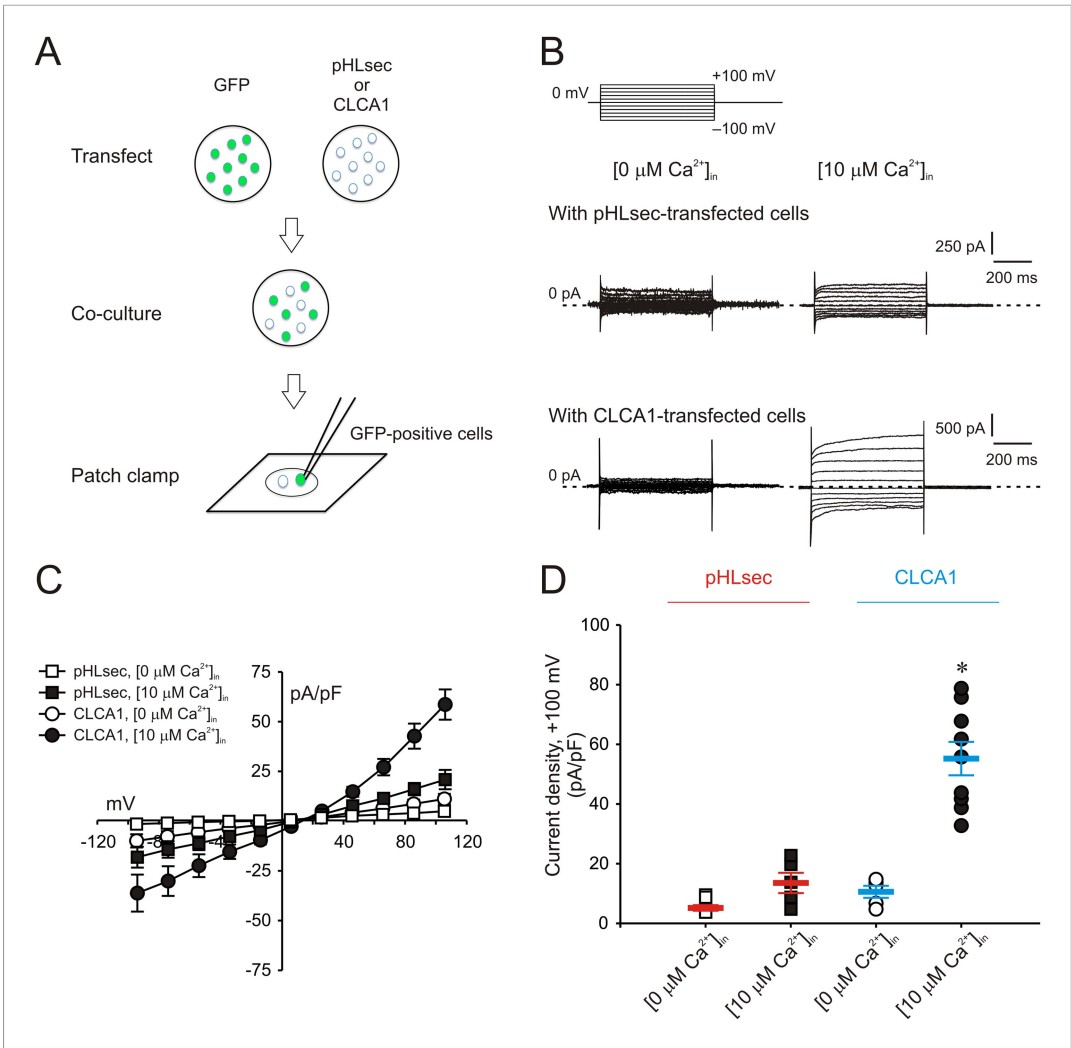

**Figure 1**. Paracrine activation of calcium-dependent chloride currents in HEK293T cells by CLCA1. (**A**) GFP-expressing cells were co-cultured with pHLsec- or CLCA1-transfected cells, and assayed for $I_{CaCC}$ by patch clamp electrophysiology. (**B–C**) Whole-cell currents measured in GFP-positive cells from experiments as in (**A**), superfused with standard extracellular solution, and in the absence or presence of 10 µM free $Ca^{2+}$ in the pipette (respectively, [0 µM $Ca^{2+}$]$_{in}$ or [10 µM $Ca^{2+}$]$_{in}$). (**B**) Representative current traces. The pulse protocol is shown at the top left. Outward currents are represented by *upward deflections*, and *dotted lines* indicate zero current. Membrane capacitance was similar in all cases at ~25 pF. (**C**) Current–voltage relationships at the end of the 600-ms voltage steps. Membrane potential values were corrected off-line for the calculated liquid junction potentials, respectively −5.5 mV ([0 µM $Ca^{2+}$]$_{in}$) and −6.0 mV ([10 µM $Ca^{2+}$]$_{in}$). Data are presented as means ± S.E. (n = 5–9). (**D**) Current density at +100 mV, from the same experiments as in (**C**). Symbols represent data from individual patches; bars indicate the means ± S.E. of all experiments. *p < 0.01 (one-way ANOVA, F = 30.3 and p = 1.2 × 10⁻⁷; followed by Tukey test).

## CLCA1 colocalizes with and increases cell surface levels of TMEM16A protein

Next, we used immunohistochemistry and confocal microscopy to examine CLCA1 and TMEM16A localization in non-permeabilized HEK293T cells. Cells transfected with pHLsec vector alone did not display noticeable staining for either CLCA1 or TMEM16A (*Figure 4A–D*), consistent with lack of endogenous expression of CLCA1 and low endogenous levels of TMEM16A in these cells (*Kunzelmann et al., 2009*; *Pritchard et al., 2014*). However, cells transfected with CLCA1 stained strongly both for CLCA1 and, surprisingly, for TMEM16A (*Figure 4E–H*), suggesting that CLCA1 increases TMEM16A protein levels. Furthermore, signal for both proteins clearly overlapped with

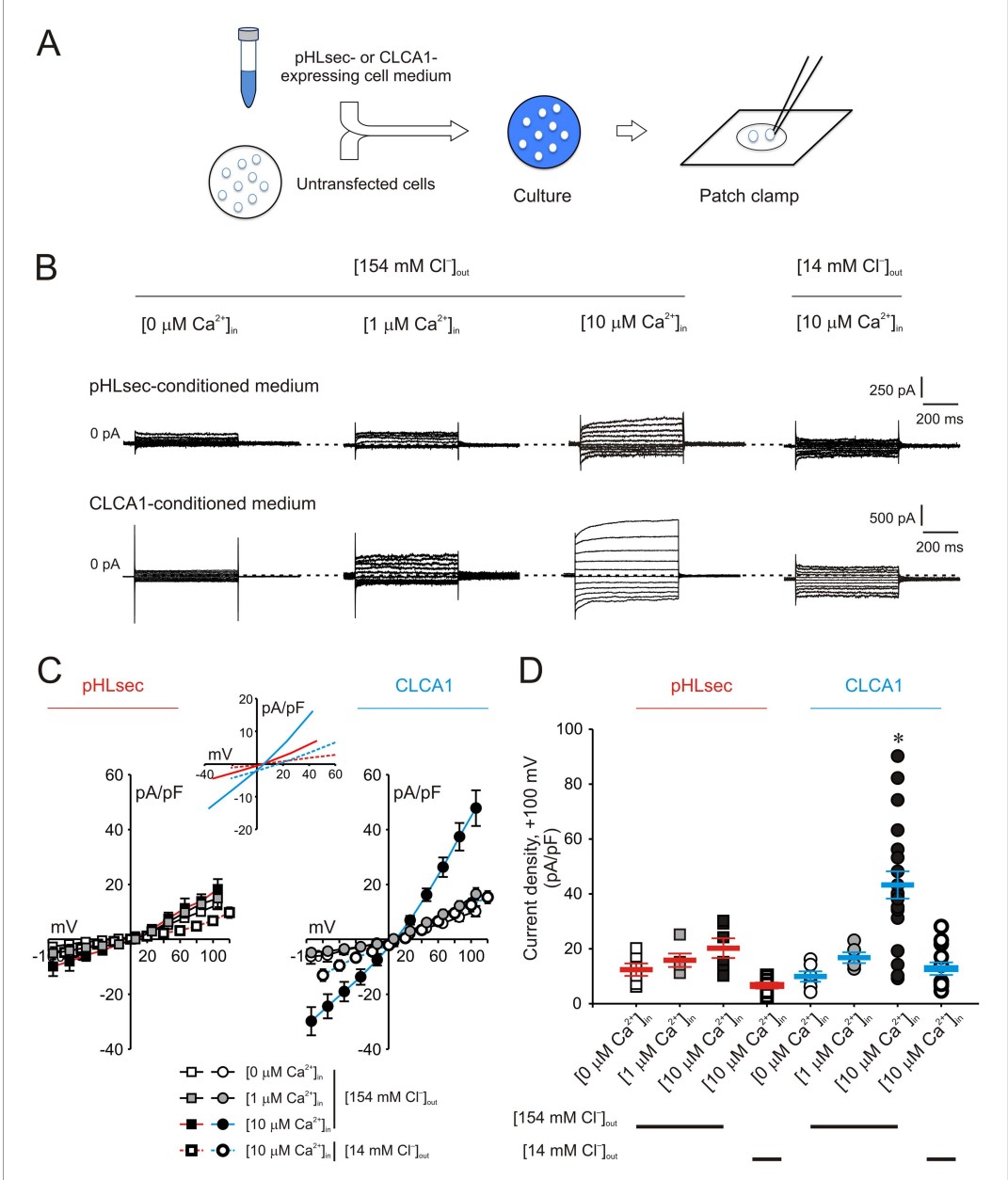

**Figure 2**. Activation of calcium-dependent chloride currents by secreted CLCA1. (**A**) Untransfected cells were cultured in medium from pHLsec- or CLCA1-expressing cells, and assayed by patch clamp electrophysiology. (**B–D**) Whole-cell currents measured in cells from experiments as in (**A**), superfused with standard ([154 mM Cl$^-$]$_{out}$) or reduced Cl$^-$ ([14 mM Cl$^-$]$_{out}$) extracellular solution; and in the absence or presence of 1 μM or 10 μM free Ca$^{2+}$ in the pipette (respectively, [0 μM Ca$^{2+}$]$_{in}$, [1 μM Ca$^{2+}$]$_{in}$ or [10 μM Ca$^{2+}$]$_{in}$). (**B**) Representative current traces obtained with the same pulse protocol and displayed as in *Figure 1B*. Membrane capacitance was similar in all cases at ~25 pF. (**C**) Current-voltage relationships at the end of the 600-ms voltage steps. Membrane potential values were corrected off-line for the calculated liquid junction potentials, respectively −5.5 mV ([0 μM Ca$^{2+}$]$_{in}$) and −6.0 mV ([1 μM Ca$^{2+}$]$_{in}$ and [10 μM Ca$^{2+}$]$_{in}$) for the experiments in [154 mM Cl$^-$]$_{out}$; and −20 mV for the experiments in [14 mM Cl$^-$]$_{out}$. Data are presented as means ± S.E. (n = 5–20). Inset, CLCA1-mediated currents right-shifted ~ +15 mV upon reduction of extracellular Cl$^-$; symbols have been removed for clarity. (**D**) Current density at +100 mV, from the same experiments as in (**C**). *Symbols* represent data from individual patches; *bars* indicate the means ± S.E. of all experiments. *p < 0.01 (one-way ANOVA, *F* = 10.4 and p = 2.1 × 10$^{-8}$; followed by Tukey test).

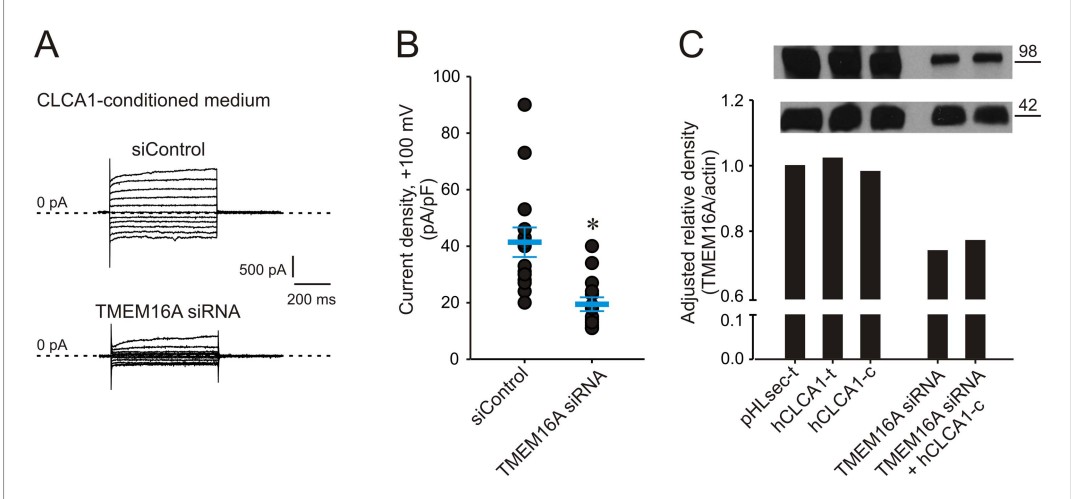

**Figure 3**. Genetic knockdown of TMEM16A inhibits CLCA1-mediated calcium-dependent chloride currents. (**A–B**) HEK293T cells transfected with RNAi negative control (siControl) or TMEM16A siRNA were incubated in CLCA1-conditioned medium and assayed by patch-clamp electrophysiology, in standard extracellular solution ([154 mM Cl⁻]_out) and 10 μM free Ca²⁺ in the pipette ([10 μM Ca²⁺]_in). (**A**) Representative current traces obtained with the same pulse protocol and displayed as in *Figure 1B*. Membrane capacitance was similar in all cases at ~25 pF. (**B**) Current density at +100 mV. Symbols represent data from individual patches (n = 14); *bars* indicate the means ± S.E. of all experiments. *p < 0.01 (unpaired Student's *t* test). (**C**) Effect of CLCA1 and/or TMEM16A siRNA treatment on TMEM16A protein expression. Upper panel: top, TMEM16A; and bottom, actin (loading control) Western blot from solubilized HEK293T cells. Lanes are labeled as follows: *pHLsec-t*, pHLsec transfected cells; *CLCA1-t*, CLCA1-transfected cells; *CLCA1-c*, cells treated with CLCA1-conditioned medium; *TMEM16A siRNA*, cells transfected with TMEM16A siRNA; *TMEM16A siRNA + CLCA1-c*, cells transfected with TMEM16A siRNA and treated with CLCA1-conditioned medium. Bar graph: quantitation of TMEM16A band intensity normalized to actin band intensity using ImageJ (NIH).

the membrane stain (WGA), consistent with a model in which CLCA1 and TMEM16A associate with and stabilize one another on the cell surface. Since secreted CLCA1 can activate TMEM16A-mediated I_CaCC in a paracrine manner (*Figures 1–3*), we carried out similar imaging experiments to determine whether exogenously applied secreted CLCA1 also increased TMEM16A surface expression. Cells cultured in media from pHLsec-transfected cells again displayed no detectable staining for either CLCA1 or TMEM16A (*Figure 4I–L*), but cells exposed to secreted CLCA1 displayed robust staining for TMEM16A. Signal for CLCA1 was also detected in a few cells, overlapping with TMEM16A and WGA staining (*Figure 4M–P*). Surprisingly, although TMEM16A surface levels increased after exposure to CLCA1, total TMEM16A in cells did not change (*Figure 3C*). These results indicate that exogenous secreted CLCA1 colocalizes with and enhances the fraction of TMEM16A located at the cell surface.

## CLCA1 associates with cell surface TMEM16A

To investigate whether CLCA1 and TMEM16A associate directly with one another on the cell surface, we adapted an assay commonly used to identify immunological receptor-ligand pairs (*Altman et al., 1996*). We previously demonstrated that CLCA1 is cut into two fragments by self-cleavage and that the N-terminal fragment is necessary and sufficient to activate CaCCs in HEK293T cells (*Yurtsever et al., 2012*). Thus, for these assays we developed a CLCA1 cell-staining reagent composed of the N-terminal fragment of CLCA1 (N-CLCA1) containing a specific biotinylation motif on the C-terminus (*Figure 5A*). Biotinylated N-CLCA1 was coupled to SA-PE (streptavidin conjugated to phycoerythrin) to produce a tetrameric fluorescent reagent with enhanced avidity toward its ligand. Cell-binding assays were carried out in the presence or absence of an anti-TMEM16A antibody raised against epitopes in the last extracellular loop and then

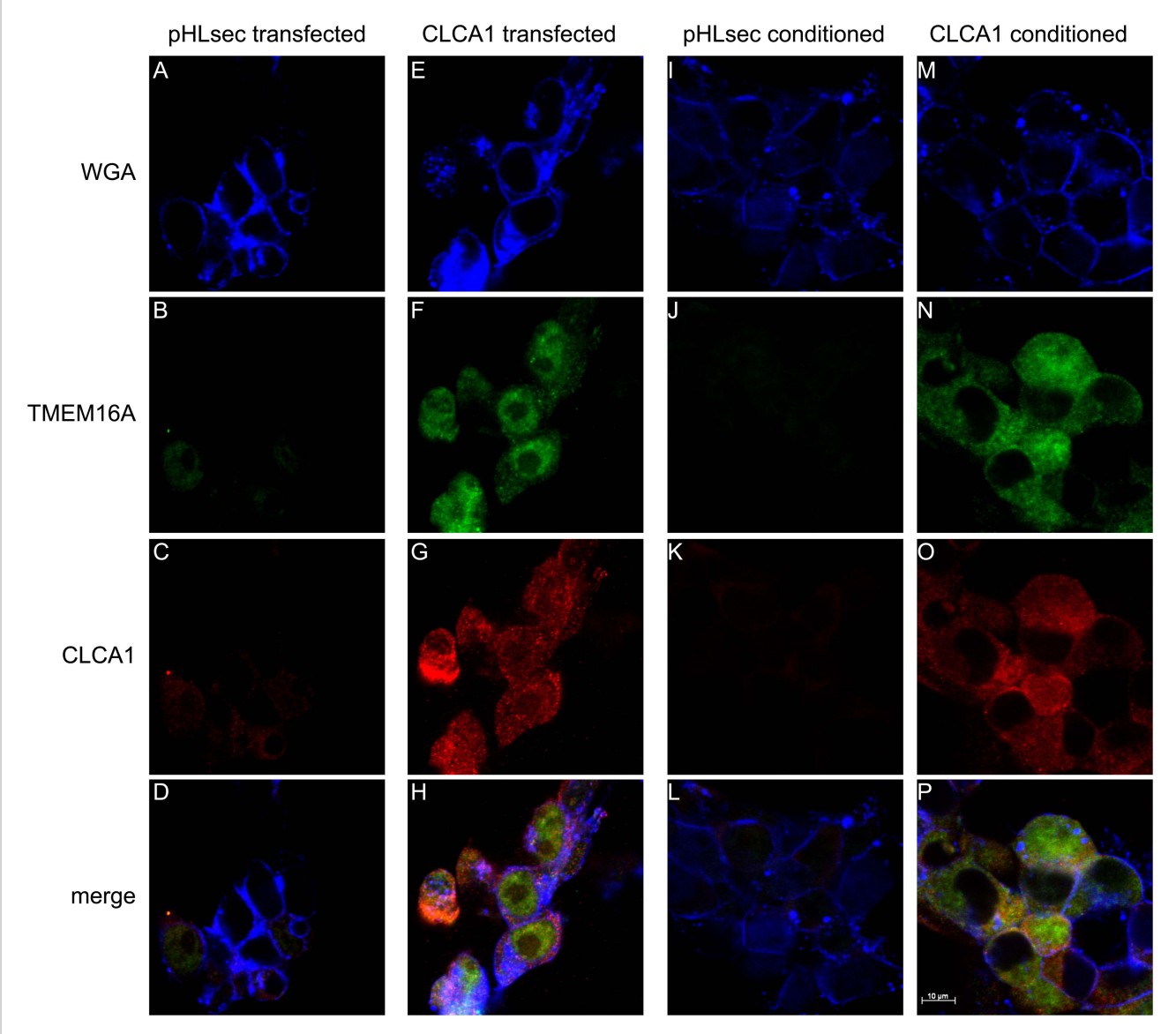

**Figure 4**. CLCA1 colocalizes with TMEM16A and increases TMEM16A surface expression. (**A**–**D**) Membrane (WGA) or immunostaining of HEK293T cells transfected with pHLsec vector; (**E**–**H**), or with CLCA1. Surface TMEM16A is greatly increased by expression of CLCA1. (**I**–**L**) Membrane (WGA) or immunostaining of HEK293T cells cultured in conditioned media from cells transfected with pHLsec vector; (**M**–**P**) or cells cultured in conditioned media from cells transfected with CLCA1. TMEM16A surface expression is greatly enhanced after cells are exposed to secreted CLCA1.

analyzed by flow cytometry. The tetramerized N-CLCA1 displayed robust binding to intact HEK293T cells compared to background, and this binding was significantly reduced by pre-incubating the cells with the anti-TMEM16A antibody (*Figure 5B*). Two control antibodies, one raised against an intracellular epitope of TMEM16A and the other an isotype control, did not affect N-CLCA1 binding (*Figure 5C*). In order to validate that the biotinylation of N-CLCA1 did not adversely affect function, we carried out whole-cell patch clamp experiments where either purified N-CLCA1 or purified biotinylated N-CLCA1 was exogenously applied to HEK293T cells. We found that both of these proteins were able to robustly activate the observed currents (*Figure 5D,E*). These results indicate that N-CLCA1 engages TMEM16A on the surface of HEK293T cells, and suggests that the enhanced TMEM16A level is a consequence of stabilization by CLCA1.

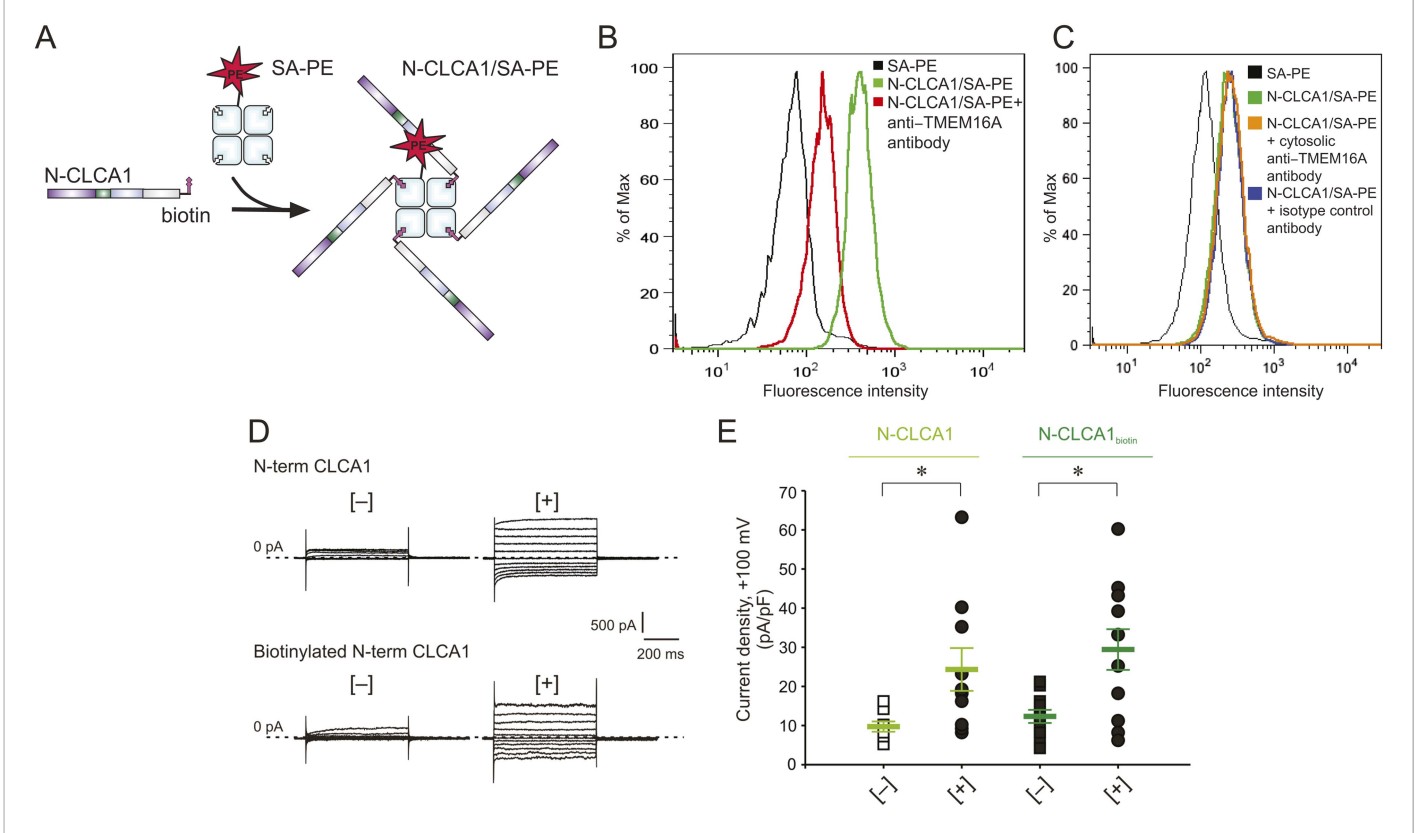

**Figure 5**. N-CLCA1 engages TMEM16A on the cell surface. (**A**) Schematic of CLCA1 N-terminal fragment (N-CLCA1) construct with specific biotinylation site and resultant tetrameric cell-staining reagent created after complexation with SA-PE. (**B**) Flow cytometry of intact HEK293T cells stained with SA-PE alone (black line), N-CLCA1/SA-PE (green line), or N-CLCA1/SA-PE in the presence of anti-TMEM16A antibody S-20 (red line). (**C**) Flow cytometry of intact HEK293T cells either stained with SA-PE alone (black line), N-CLCA1/SA-PE (green line), N-CLCA1/SA-PE in the presence of anti-TMEM16A antibody C-5 (raised against an intracellular TMEM16A epitope; orange line), or N-CLCA1/SA-PE in the presence of anti-Aquaporin5 antibody G-19 (blue line). (**D–E**) Cells were incubated in the absence ([–]) or presence ([+]) of purified N-terminal (N-term) CLCA1 protein before (N-CLCA1) or after biotinylation (N-CLCA1$_{biotin}$), and assayed by patch-clamp electrophysiology, in standard extracellular solution ([154 mM Cl$^-$]$_{out}$) and 10 μM free Ca$^{2+}$ in the pipette ([10 μM Ca$^{2+}$]$_{in}$). (**D**) Representative current traces obtained with the same pulse protocol and displayed as in *Figure 1B*. Membrane capacitance was similar in all cases at ~25 pF. (**E**) Current density at +100 mV. *Symbols* represent data from individual patches (n = 8–11); *bars* indicate the means ± S.E. of all experiments. *p < 0.05 (unpaired Student's *t* test).

## Discussion

It has been demonstrated that purified TMEM16A protein reconstituted in proteoliposomes recapitulates the permeation, pharmacological, voltage- and Ca$^{2+}$-dependence properties of TMEM16A channels characterized in heterologous expression systems and native cell models (*Terashima et al., 2013*), which implies that TMEM16A does not require association with other proteins for CaCC activity. However, a recent proteomics approach has identified a large number of endogenous proteins implicated in protein trafficking, surface expression, folding and stability that interact with TMEM16A, including SNAREs such as syntaxins and syntaxin-binding proteins, and the ezrin-radixin-moesin (ERM) scaffolding complex (*Perez-Cornejo et al., 2012*). Our data identify CLCA1 as the first secreted direct regulator of TMEM16A, and our findings suggest that CLCA1 may modulate TMEM16A channel activity by stabilizing it at the cell surface, much like the SNARE and ERM protein networks of the TMEM16A interactome.

So how does CLCA1 modulate TMEM16A currents? We observe that CLCA1 increases TMEM16A surface expression without increasing expression of the protein (*Figure 3C*). A model consistent with our data and the current literature is that CLCA1 engages and stabilizes dimeric TMEM16A on the surface of the cell. Previous studies have shown that TMEM16A can exist as a dimer (*Fallah et al., 2011*; *Sheridan et al., 2011*), dimerization being mediated by an

intracellular region in the N-terminus of TMEM16A (*Tien et al., 2013*). Mutations to this region abolish dimerization, prevent protein trafficking to the plasma membrane, and, consequently, ablate channel activity (*Tien et al., 2013*). These observations indicate that dimerization regulates TMEM16A trafficking and channel activity. It is possible that TMEM16A dynamically shuttles between the cell surface and intracellular compartments. One possibility is that CLCA1 engages monomeric TMEM16A and drives dimerization; alternatively, CLCA1 may engage and stabilize dimeric TMEM16A on the cell surface, thereby preventing its removal and, consequently, increasing calcium-dependent chloride currents (*Figure 6*). This highlights an unprecedented mechanism for regulating ion channel currents by a secreted protein.

The identification of CLCA1 as a modulator of TMEM16A activity raises the possibility of functional associations between other CLCA and TMEM16 family members. Four to eight CLCA (*Patel et al., 2009*) and ten TMEM16 family members (*Pedemonte and Galietta, 2014*) are expressed in mammalian tissues. A number of these TMEM16 proteins have poorly defined functions and do not obviously traffic to the cell surface when expressed alone (*Duran et al., 2012*). Future studies will be needed to determine whether other CLCA proteins can associate with other TMEM16 proteins and influence their function. CLCA1 (*Yang et al., 2013*), CLCA2 (*Sasaki et al., 2012*; *Walia et al., 2012*), and CLCA4 (*Yu et al., 2013*) have all been implicated in various cancers as have a number of TMEM16 proteins (*West et al., 2004*; *Dutertre et al., 2010*; *Duvvuri et al., 2012*; *Liu et al., 2012*; *Qu et al., 2014*), and such studies could have tremendous implications for cooperative CLCA/TMEM16 roles in cancer and other diseases.

Here we report that secreted CLCA1 modulates TMEM16A-dependent $I_{CaCC}$, and that this activation can occur in a paracrine fashion. Furthermore, we show that CLCA1 and TMEM16A colocalize and physically interact on the surface of mammalian cells, with CLCA1 increasing the level of TMEM16A protein at the cell surface. We thus demonstrate a first downstream target of CLCA proteins, solving the 20-year-old mystery regarding how CLCA proteins activate $I_{CaCC}$, and provide the first example of a secreted protein modulator of TMEM16A activity. CLCA1 (*Alevy et al., 2012*) and TMEM16A (*Huang et al., 2012*; *Scudieri et al., 2012*) have been separately observed to play critical roles in chronic inflammatory airway disease models. Our findings have significant implications for the roles of CLCA1 and TMEM16A proteins as cooperative partners, not only in the physiology and pathophysiology of the airways, but also in those of other tissues and organs.

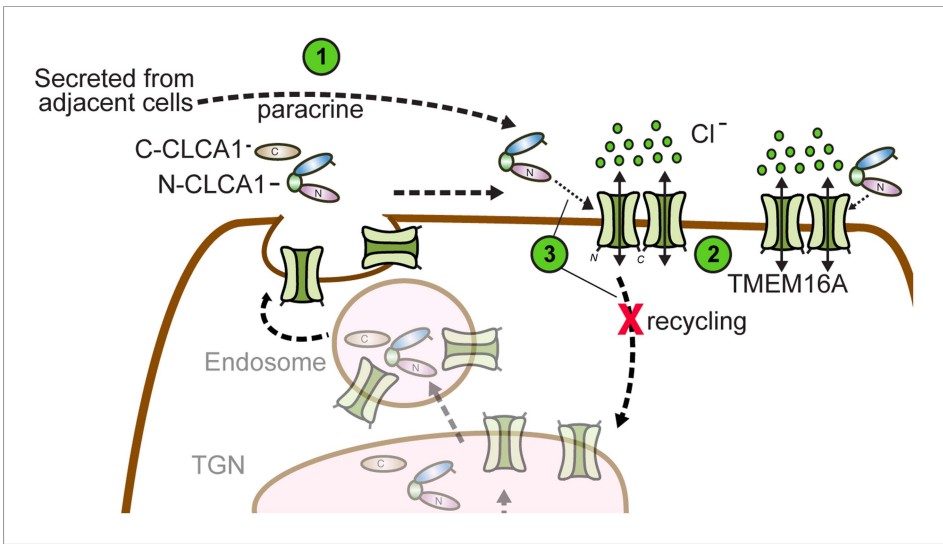

**Figure 6**. Model for CLCA1 modulation of TMEM16A-mediated calcium-dependent chloride currents. Following secretion and self-cleavage of CLCA1, the N-terminal fragment (N-CLCA1) acts in paracrine fashion (1). Dimerization appears to regulate surface trafficking of TMEM16A. N-CLCA1 engages TMEM16A on the cell surface (2), stabilizing TMEM16A dimers, preventing internalization (3) and in turn, results in increased TMEM16A surface expression and calcium-dependent chloride current density.

## Materials and methods

### Reagents

The following commercial antibodies were used according to the manufacturer's specifications: mouse-anti-human-TMEM16A monoclonal antibody C-5 (Santa Cruz Biotechnology, Dallas, TX); goat-anti-human-TMEM16A polyclonal antibody S-20 (Santa Cruz Biotechnology); mouse-anti-actin monoclonal antibody C4 (Millipore, Billerica, MA), goat-anti-human-Aquaporin5 polyclonal antibody G-19 (Santa Cruz Biotechnology); rabbit anti-6-His-antibody-HRP conjugate (Bethyl Laboratories, Montgomery, TX), goat anti-mouse IgG antibody-HRP conjugate (Santa Cruz Biotechnology); wheat germ agglutinin (WGA)-Alexa Fluor 633 conjugate (Life Technologies, Carlsbad, CA); donkey anti-goat IgG-Alexa Fluor 488 conjugate (Life Technologies); donkey anti-rabbit IgG-Alexa Fluor 594 conjugate (Life Technologies); and rabbit anti-human CLCA1 polyclonal antibody 1228 (Biosystems, Rockford, IL). Mouse anti-human CLCA1 monoclonal antibody 8D3 was produced in-house and used as previously described (*Alevy et al., 2012*; *Yurtsever et al., 2012*). Streptavidin conjugated to phycoerythrin (SA-PE) was purchased from BD Biosciences (San Jose, CA). Hype-5 transfection reagent was purchased from OZ Biosciences (San Diego, CA).

### Heterologous expression of CLCA1

Full length human CLCA1 (22–914) (CLCA1) cloned into pHLsec vector was used throughout (*Yurtsever et al., 2012*). HEK293T cells were cultured in 6-well dishes in Dulbecco's Modified Eagle Medium (Life Technologies) supplemented with 10% fetal bovine serum, $10^5$ units/l penicillin and 100 mg/l streptomycin, at 37°C and 5% $CO_2$. Cells were transfected at 80% confluency using 293fectin transfection reagent (Life Technologies) at a 1:2 ratio (µg DNA: µl 293fectin), using 1 µg of plasmid DNA per 1 million cells. Experiments were conducted in cells that were transiently transfected with CLCA1, or in cells that were exposed to exogenous CLCA1 protein via means of two different experimental approaches: either co-culture with CLCA1 transfected cells; or treatment with CLCA1-conditioned medium. For co-culture experiments, cells transfected with CLCA1, empty pHLsec vector (pHLsec), or EGFP-pCDNA3.1 plasmid (GFP) were trypsinized 24 hr post-transfection, and GFP-expressing cells were mixed at a 1:1 ratio with either CLCA1 or pHLsec-transfected cells, and replated at low density on UV-sterilized, 8 mm round German glass coverslips (Electron Microscopy Sciences, Hatfield, PA). Following trypsin treatment, all transfected cells were pelleted by centrifugation and washed with sterile PBS prior to replating to prevent carry-over of transfection medium. After 24 hr, the GFP-expressing cells were assayed for $Ca^{2+}$-dependent $Cl^-$ currents by patch clamp electrophysiology. For conditioned medium experiments, cells were transfected with either CLCA1 or pHLsec for 6 hr, then transfection medium was removed, cells were washed with sterile PBS, and fresh medium was applied; following overnight incubation, medium from these cells was harvested and centrifuged gently (1500×*g*, 5 min) to remove non-adherent cells. Untransfected cells were plated onto round coverslips and incubated for 24 hr in 2 ml of cleared CLCA1- or pHLsec-conditioned medium supernatants.

### Recombinant expression of CLCA1 and in vitro biotinylation

The N-terminal fragment of CLCA1 (22–694; N-CLCA1) was cloned into pHL-Avitag3 vector (*Aricescu et al., 2006*), which contains a BirA biotin ligase recognition motif and hexahistidine tag at the C-terminus. This secreted protein was expressed in 293F cells via transient transfection using Hype-5 at 1:1.5 µl ratio (µg DNA: µl Hype-5), using 1 µg of plasmid DNA per 1 million cells. Media supernatants were harvested after 72 hr. Protein was purified from media supernatant using Ni-NTA chromatography and eluted in 5 ml buffer A (50 mM $K_2HPO_4$ pH 8, 300 mM NaCl and 250 mM imidazole). Purified N-CLCA1 was concentrated to a final volume of 300 µl in a centrifuge concentrator and protein concentration was calculated from absorbance at 280 nm. For the experiments reported in *Figure 5D,E*, protein was added onto the untransfected cells at 10 µg/ml and incubated for 24 hr prior to whole-cell patch clamp experiments. The same volume of buffer A was added onto cells as buffer control. For in vitro biotinylation, N-CLCA1 containing the specific biotinylation tag at the C-terminus was exchanged into buffer B (100 mM Tris pH 7.5, 200 mM NaCl, and 5 mM $MgCl_2$) and specifically biotinylated by addition of biotin and *Escherichia coli* BirA ligase (produced and purified in-house) at 4°C overnight. Excess biotin was removed using a 2 ml desalting column.

Biotinylated N-CLCA1 was added onto the untransfected cells at 10–50 µg/ml and incubated for 24 hr prior to whole-cell patch clamp experiments. The same volume of buffer B was added onto cells as buffer control.

## siRNA knockdown of TMEM16A

To investigate the molecular identity of CLCA1-modulated CaCCs, a targeted approach was taken focusing on TMEM16A. For siRNA knockdown of TMEM16A, cells plated in 48-well plates were transfected with either 200 nM TMEM16A siRNA (HSS123904; 5′-AAG UUA GUG AGG UAG GCU GGG AAC C-3′, Life Technologies) or 200 nM medium GC-content Stealth RNAi negative control (12935300; 5′-GGU UCC CAG CCU ACC UCA CUA ACU U-3′, Life Technologies) using Lipofectamine 2000 (Life Technologies) at a 20:2 ratio (pmol siRNA: µl Lipofectamine 2000); 24 hr later, cells were plated onto round coverslips and incubated for an additional 24 hr in CLCA1- or pHLsec-conditioned medium as described above. *TMEM16A* knockdown was estimated at 60–70% as assayed by qPCR.

## Whole-cell patch clamp recordings

Experiments were performed at 25°C, 24 hr after co-culture or incubation in conditioned medium. Micropipettes were prepared from non-heparinized hematocrit glass (Kimble-Chase, Vineland, NJ) on a horizontal puller (Sutter Instruments, Novato, CA), and filled to a typical electrode resistance of 2 megaohms with pipette solution containing 150 mM N-methyl-D-glucamine (NMDG) chloride, 10 mM Hepes, 2 mM $MgCl_2$, 8 mM HEDTA, and 5.8 mM $CaCl_2$ to attain 10 µM free $Ca^{2+}$, as calculated by means of the CaBuf program (available through Katholieke Universiteit Leuven). Selected experiments were performed with a pipette solution containing (mM) 150 NMDG chloride, 10 Hepes and 2 $MgCl_2$, in the absence ([0 µM $Ca^{2+}$]$_{in}$) or presence of 5 mM EGTA and 4 mM $CaCl_2$ to attain 1 µM free $Ca^{2+}$ ([1 µM $Ca^{2+}$]$_{in}$). The pH of all pipette solutions was adjusted to 7.1 with Tris. The bath solution was 10 mM Hepes, 1 mM $CaCl_2$ and 1 mM $MgCl_2$; plus 150 mM NaCl (standard extracellular, [154 mM $Cl^-$]$_{out}$), or 140 mM Na-gluconate and 10 mM NaCl (reduced extracellular $Cl^-$ [14 mM $Cl^-$]$_{out}$), and adjusted to pH 7.4 with Tris. After formation of a gigaohm seal and establishment of whole-cell configuration, cells were voltage-clamped at 0 mV. A pulse protocol was applied in which membrane potential was held at 0 mV for 50 ms and stepped to a test value for 600 ms before returning to the holding potential for an additional 400 ms. The test potential varied from −100 to +100 mV in 20 mV increments. Membrane capacitance was calculated from the integral of the current transient in response to 10 mV depolarizing pulses, and was monitored for stability throughout the experiment. Data were filtered at 2 kHz, and signals were digitized at 5 kHz with a Digidata 1322A (Molecular Devices, Sunnyvale, CA). MultiClamp 700B Commander and pClamp software (Molecular Devices) were used for pulse protocol application and data acquisition. Data were analyzed using Clampfit 10.1 (Molecular Devices). Liquid junction potentials were calculated using Clampex JPCalc software and command voltages were corrected a posteriori as specified in the figure legends. Results are presented as mean ± S.E., differences between two groups were assessed by unpaired Student's *t* test with Welch's correction, and differences between more than two groups were evaluated by one-way ANOVA and *post-hoc* Tukey test (Prism 5.0c, GraphPad Software, San Diego, CA).

## Immunohistochemistry

For staining experiments, cells were either transfected or exposed to conditioned medium as described above. Following 24 hr incubation, cells were fixed on glass slides with 4% paraformaldehyde (PFA) in PBS for 5 min and washed twice with PBS. Cells were blocked for 1 hr at room temperature with 1% blocking solution in PBS (Life Technologies) and then incubated with primary antibodies (rabbit anti-human CLCA1 polyclonal antibody 1228 at 1:100 dilution and goat-anti-human-TMEM16A polyclonal antibody S-20 at 1:50 dilution) overnight at 4°C. Slides were washed and incubated with WGA-Alexa Flour 633 conjugate (5 µg/ml) for 30 min at room temperature, followed by secondary antibodies (donkey anti-rabbit IgG-Alexa Fluor 594 conjugate at 1:250 dilution and donkey anti-goat IgG-Alexa Fluor 488 conjugate at 1:200 dilution) for 2 hr at room temperature. Washed slides were then mounted in VECTASHIELD H-1200 Mounting Medium with DAPI (Vector Laboratories, Burlingame, CA). Confocal microscopy was carried out using a Zeiss LSM 510 META Confocal Laser Scanning Microscope

(Carl Zeiss Microscopy, Thornwood, NY). The images were acquired with LSM 4.2 software and batch processed with AxioVision 4.8.2 (Carl Zeiss Microscopy).

## Flow cytometry binding assay

Human CLCA1 was assayed for binding to cell-surface TMEM16A using a flow cytometry-based binding assay. Prior to staining, intact HEK293T cells were treated with human FcR blocking reagent at 1:100 dilution (Miltenyi Biotec, San Diego, CA) for 15 min. The biotinylated N-CLCA1 was pre-incubated with SA-PE at a 4:1 molar ratio for 15 min at room temperature to produce fluorescently labeled tetramers of N-CLCA1 (N-CLCA1/SA-PE). Cells ($4 \times 10^5$ cells/sample) were either stained with SA-PE alone (1:50) or N-CLCA1/SA-PE (1:50) diluted in PBS containing 1% BSA (FACS buffer) at 4°C. In order to validate specific binding of N-CLCA1/SA-PE to cell surface TMEM16A, goat-anti-human-TMEM16A polyclonal antibody S-20 (1:10), which was raised against a 15–20 amino acid peptide within residues 820–870 (corresponding to the last extracellular loop; UniProt Q5XXA6), was added prior to addition N-CLCA1/SA-PE. A goat polyclonal IgG antibody for human Aquaporin5 was used (1:10) as an isotype control for the blocking antibody S-20. Mouse-anti-human-TMEM16A monoclonal antibody C5, which binds a cytosolic epitope, was used as a second control antibody (1:10). Following staining, cells were washed with FACS buffer, and then analyzed by flow cytometry (BD FACScan). Data analysis was performed using FlowJo (Tree Star, Ashland, OR).

## Western blotting and densitometric quantitation

HEK293T cells were pelleted, lysed in lysis buffer (1.5 mM $KH_2PO_4$, 2.7 mM KCl, 4.3 mM $Na_2HPO_4$, 137 mM NaCl and 1% Triton X-100 in deionized water), and then diluted 1:2 in 2× SDS containing 2-mercaptoethanol. Samples were boiled for 5 min, and then loaded on a 4–12% bis-tris Nupage gel (Life Technologies). The proteins were transferred to nitrocellulose membranes using an iBlot Gel Transfer Device (Life Technologies). Membranes were blocked by 0.5% nonfat milk in PBS with 0.1% TWEEN. Primary antibodies (mouse α-human CLCA1 8D3, 1:4000; mouse-anti-human-TMEM16A monoclonal antibody C-5, 1:1000; mouse-anti-actin monoclonal antibody C4, 1:5000) in blocking buffer were incubated on the membrane for 15 min. Following three washes with PBS-TWEEN, secondary antibodies (goat-anti-mouse IgG-HRP conjugate 1:5000) in blocking buffer were applied for 15 min. After three PBS-TWEEN washes, signal was detected using Pierce ECL Western Blotting Substrate (Thermo Fisher Scientific, Rockford, IL). Developed films were scanned and converted to 8-bit tiff files. Protein bands were processed equally and the pixel intensities were quantified with ImageJ 1.48 (http://imagej.nih.gov/ij).

## Acknowledgements

This work was supported by NIH R01-HL119813 (to TJB) and R01-HL54171 (to CGN), American Lung Association RG-196051 (to TJB), a CIMED Pilot and Feasibility grant (to TJB), and American Heart Association Predoctoral Fellowship PRE19970008 (to ZY).

## Additional information

### Funding

| Funder | Grant reference number | Author |
| --- | --- | --- |
| National Heart, Lung, and Blood Institute (NHBLI) | R01-HL119813 | Tom J Brett |
| American Heart Association (AHA) | PRE19970008 | Zeynep Yurtsever |
| American Lung Association | RG-196051 | Tom J Brett |
| National Heart, Lung, and Blood Institute (NHBLI) | R01-HL54171 | Colin G Nichols |
| Washington University in St. Louis | Pilot and Feasibility Grant | Tom J Brett |

The funders had no role in study design, data collection and interpretation, or the decision to submit the work for publication.

## Author contributions
MS-R, ZY, Conception and design, Acquisition of data, Analysis and interpretation of data, Drafting or revising the article; CGN, TJB, Conception and design, Analysis and interpretation of data, Drafting or revising the article

## Author ORCIDs
Tom J Brett, http://orcid.org/0000-0002-6871-6676

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
