## [Decision Letter]

Thank you for sending your work entitled “Secreted CLCA1 modulates TMEM16A to activate Ca^2+^-dependent chloride currents in human cells” for consideration at *eLife*. Your article has been favorably evaluated by Michael Marletta (Senior editor) and 3 reviewers, one of whom, Richard Aldrich, is a member of our Board of Reviewing Editors.

The Reviewing editor and the other reviewers discussed their comments before we reached this decision, and the Reviewing editor has assembled the following comments to help you prepare a revised submission.

They found the work to be interesting and potentially quite important. However there remains some necessary additional work to validate the conclusions and to rule out alternative interpretations of the results. Satisfactory completion of the following experiments will be necessary for further consideration. They should be relatively easily completed with existing approaches.

CLCA proteins are a family of soluble proteins that activate Ca-dependent chloride currents in mammalian cells. The mechanism by which this occurs has been unknown. In this work, the authors conclude that secreted CLCA1 binds directly to TMEM16a channels to stabilize them at the plasma membrane. This conclusion is of interest because it identifies the molecular target of CLCA1 (TMEM16a) and proposes a mechanism by which the activation of TMEM16a occurs. The proposed mechanism would likely be of relevance to many other CLCA and TMEM family members.

The proposed mechanism by which TMEM16a is activated is suggested by a combination of two types of experiments:

(1) Functional assays: Endogenous Ca-dependent chloride currents are activated by co-culturing HEK293T cells with CLCA1-expressing cells or by culturing HEK293T cells with extracellular media obtains from cells expressing CLCA1

(2) Co-localization and binding assays: (2a) Antibody staining shows colocalization of CLCA1 AND TMEM16A, and (2b) a fluorescent biotinylated N-CLCA1 construct interacts with TMEM16a-expressing cells as detected by flow cytometry.

1) Upon viewing the first set of experiments, we wondered whether there might be other explanations for the activation of Ca-dependent chloride currents, perhaps something other than the CLCA1 secreted by the cells is responsible for activating the currents. (Or, less interestingly, perhaps there is an artifact from incomplete washing of transfection reagents.) It would be straightforward for the authors to alleviate these concerns, simply show that direct addition of the CLCA1 protein is sufficient to activate the currents. In previous work, the authors have demonstrated their ability to express and purify CLCA1 (41).

2) Further, in this work, the binding experiments were done with a biotinylated CLCA1 variant. To be certain that this binding is relevant to activation, it should be shown that this variant not only binds but also induces the expected functional result (i.e. activation of TMEM16a currents). These experiments can be done with existing reagents using methods (patch-clamp recording) already performed in this study. If CLCA1 protein cannot be shown to activate currents, then the model presented for the mechanism of activation is not supported.

3) The identification of TMEM16a channels as the molecular target for CLCA1 is reasonable based on the properties of the currents and the siRNA experiments, but the experiments with inhibitors are not convincing and should either be improved or eliminated from the manuscript. The inhibitors used are said to be “potent” but are used at high concentration (10 uM). At such concentrations off-target effects become likely. This is particularly of concern because of the method by which inhibition was detected: cells incubated in inhibitor were compared to cells that were not incubated with inhibitor. A more convincing experiment would be to measure currents before, after, and upon washout of inhibitor (at a reasonable concentration).

4) The authors present only a cursory analysis of the properties of the complex's currents under rather “extreme” conditions (+100 mV, high internal Cl-, high internal Ca^2+^ and long incubations with inhibitors). For example, is the CLCA1-induced current increase the same at all voltages and/or at all Ca^2+^ concentrations?

---

## [Author Response]

*1) Upon viewing the first set of experiments, we wondered whether there might be other explanations for the activation of Ca-dependent chloride currents, perhaps something other than the CLCA1 secreted by the cells is responsible for activating the currents. (Or, less interestingly, perhaps there is an artifact from incomplete washing of transfection reagents.) It would be straightforward for the authors to alleviate these concerns, simply show that direct addition of the CLCA1 protein is sufficient to activate the currents. In previous work, the authors have demonstrated their ability to express and purify CLCA1 (*[41]*)*.

Care was taken to remove the transfection mixtures in these experiments, and we have now more explicitly stated that in the Methods section. In addition, we have carried out the suggested experiment (exogenously applying purified N-CLCA1 to 293T cells) and have demonstrated that this purified protein is capable of activating the calcium-activated chloride currents we observed in these cells (Figure 5).

*2) Further, in this work, the binding experiments were done with a biotinylated CLCA1 variant. To be certain that this binding is relevant to activation, it should be shown that this variant not only binds but also induces the expected functional result (i.e. activation of TMEM16a currents). These experiments can be done with existing reagents using methods (patch-clamp recording) already performed in this study. If CLCA1 protein cannot be shown to activate currents, then the model presented for the mechanism of activation is not supported*.

We have performed the suggested patch-clamp experiments using the purified, biotinylated N-CLCA1 added exogenously to the 293T cells. This biotinylated protein activates the calcium-activated chloride currents just as robustly as the non-biotinylated protein (Figure 5), indicating that the biotinylated N-CLCA1 is functional and, furthermore, supporting our presented model.

*3) The identification of TMEM16a channels as the molecular target for CLCA1 is reasonable based on the properties of the currents and the siRNA experiments, but the experiments with inhibitors are not convincing and should either be improved or eliminated from the manuscript. The inhibitors used are said to be “potent” but are used at high concentration (10 uM). At such concentrations off-target effects become likely. This is particularly of concern because of the method by which inhibition was detected: cells incubated in inhibitor were compared to cells that were not incubated with inhibitor. A more convincing experiment would be to measure currents before, after, and upon washout of inhibitor (at a reasonable concentration)*.

As per the reviewers’ suggestion, we have removed the inhibitor data from this manuscript and plan to perform more comprehensive experiments with these inhibitors in a future manuscript. Because these experiments have been removed, we have removed one author from the manuscript (Arthur Romero), as his sole contribution was the synthesis of one of the inhibitors (MONNA).

*4) The authors present only a cursory analysis of the properties of the complex's currents under rather “extreme” conditions (+100 mV, high internal Cl-, high internal Ca*^*2+*^
*and long incubations with inhibitors). For example*, *is the CLCA1-induced current increase the same at all voltages and/or at all Ca*^*2+*^
*concentrations?*

We have addressed this by performing a more detailed assessment of the biophysical properties of the CLCA1-dependent currents, and our manuscript now includes experiments in physiological and reduced extracellular Cl^–^ concentrations, and in the presence of 0, 1 or 10 μM free pipette Ca^2+^ (Figure 2). As suggested by the reviewers, we have eliminated the experiments with inhibitors (see reply to Comment #3 above). Also, we are now displaying the current-voltage relationships in the full experimental range (Figures 1 and 2), in addition to highlighting the changes in maximum current density (pA/pF at +100 mV, Figures 1 and 2), and the right shift in reversal potential by decreasing extracellular chloride. We believe that our data robustly show that CLCA1-mediated currents display essentially the same Ca^2+^-, voltage- and Cl^–^ dependence as those described in the literature for Ca^2+^-activated Cl^–^ channels, and in particular TMEM16A.